# A Novel Clinical Research Modality for Enrolling Diverse Participants Using a Diverse Team

**DOI:** 10.3390/brainsci10070434

**Published:** 2020-07-08

**Authors:** Phoebe Lay, Tapasvini Paralkar, Syed Hadi Ahmed, Minha Ghani, Sara Muneer, Ramsha Jinnah, Carolyn Chen, Jack Zeitz, Alejandra Nitsch, Nico Osier

**Affiliations:** 1College of Natural Sciences, The University of Texas at Austin, Austin, TX 78712, USA; phoebelay97@gmail.com (P.L.); carolynchen98@gmail.com (C.C.); jackzeitz97@gmail.com (J.Z.); 2College of Liberal Arts, The University of Texas at Austin, Austin, TX 78712, USA; tapasvini@gmail.com (T.P.); minha.ghani@gmail.com (M.G.); 3McCombs School of Business, The University of Texas at Austin, Austin, TX 78712, USA; hadi17.apple@gmail.com (S.H.A.); jinnahramsha23@gmail.com (R.J.); allienitsch2@gmail.com (A.N.); 4Dell Medical School, The University of Texas at Austin, Austin, TX 78712, USA; sara.muneer90@live.com; 5Dell Medical School, The University of Texas at Austin School of Nursing, Austin, TX 78712, USA

**Keywords:** brain injuries, traumatic brain injury, children, diversity, student-led, participant-focused, recruitment, sample, methods

## Abstract

The advancement of the pediatric traumatic brain injury (TBI) knowledge base requires biospecimens and data from large samples. This study seeks to describe a novel clinical research modality to establish best practices for enrolling a diverse pediatric TBI population and quantifying key information on enrollment into biobanks. Screening form responses were standardized and cleaned through Google Sheets. Data were used to analyze total individuals at each enrollment stage. R was utilized for final analysis, including logistic model and proportion statistical tests, to determine further significance and relationships. Issues throughout data cleaning shed light on limitations of the consent modality. The results suggest that through a diverse research team, the recruited sample exceeds traditional measures of representation (e.g., sex, race, ethnicity). Sex demographics of the study are representative of the local population. Screening for candidates is critical to the success of the consent modality. The consent modality may be modified to increase the diversity of the study population and accept bilingual candidates. Researchers must implement best practices, including increasing inclusivity of bilingual populations, utilizing technology, and improving participant follow-up, to improve health disparities for understudied clinical populations.

## 1. Introduction

Traumatic brain injury (TBI) is defined as an alteration in brain function, or other evidence of brain pathology, caused by an external force [1]. Common causes of TBI include falls, motor vehicle collisions (MVC), and contact sports. Adults and children are both susceptible to deficits after TBI, but these symptoms are often not addressed until the individual returns to their health care provider [2]. If left unresolved, TBI can have cognitive, physiological, social, and emotional impact on daily functioning and future health. TBI also impacts family responsibilities and impairs the ability to work through changes in emotion, mood, and personality such as psychological distress [3]. A better understanding of TBI health outcomes can mitigate the aforementioned impacts on a patient’s life.

Children sustaining complicated mild TBI have been found to be more vulnerable to developing mild neuropsychological dysfunctions that persist chronically, when compared to adults [4]. Despite the implications of these findings, both pre-clinical [5] and clinical [6] research addressing pediatric TBI is limited. A 2018 systematic review found that despite increasing research, gaps remain in the diagnosis, prognosis, and management of pediatric mild TBI [7]. Specifically, current evidence-based guidelines are insufficient and clinical trials have failed [8]. Advancing the pediatric TBI knowledge base and our capacity to diagnose and treat these patients will require generation of significantly more clinical data and the development of large biospecimen repositories for this population.

Enrollment rates in TBI studies can be studied systematically, including the reasons why candidates ultimately opt-out of recruitment [9]. The decision to enroll in a pediatric research study is ultimately made by the participant and their parent or legal authorized representative (LAR), hereafter referred to as LAR. Low enrollment rates may be a result of several barriers, including: a relatively small number of participants who meet inclusion criteria, ethical concerns regarding the participation of children in research, LAR hesitation in allowing their child to participate [10], and consenter unavailability. Additionally, a consent protocol that does not consider variation in cultural values and understanding of the research study can unintentionally lower participation and retention rates, as well as decrease the sense of trust between communities and scientific researchers [11]. A risk–benefit assessment, influenced by the pre-existing perceptions of consent by the LAR, may also result in a lower likelihood of enrollment [12]. Enrollment can be facilitated by improving the approach to consenting participants [9]. According to a literature review that explores enrollment in pediatric clinical trials, a limiting factor for participation is the level of study understanding by the participant and LAR. The review suggests that enrollment rates can thus be enhanced through effective communication, open discussion, and delivery of information in all languages spoken by the target study population [13]. Schnur et al. [9] discuss a protocol that includes pre-screening of the census and screening of candidate’s medical charts; this technique impacted enrollment rates by minimizing the time spent approaching ineligible cases and effectively using time with the candidates as the consenter is already well-informed. Notably, this study of best enrollment practices was conducted using data from an intensive care unit (ICU), with candidates approached after medical stabilization. Important differences exist between the ICU and emergency department (ED) that could impact enrollment. For example, TBI cases can enter the ED at any time of day with very brief visit durations, so enrollment rates are often reduced due to rapid turnover. Therefore, many studies maximize participant approaches by employing full-time clinical research coordinators. However, early stage investigators may lack the funds to have such personnel available 24/7. There remains an important gap regarding how to maximize recruitment of ED patients into pediatric TBI biobanks for investigators with limited funding. This paper draws from our past work [9] and seeks to describe a new student-led, participant-focused approach to consenting a diverse sample of pediatric patients and to quantify key information about enrollment into our biobank using this modality.

## 2. Materials and Methods

Following institutional review board (IRB) approval, the study recruited 93 children between 5 to 16 years of age into a biobank. The sample included 33 pediatric TBI cases and 60 orthopedic controls. The recruitment period occurred between June 2018 and November 2019. Unlike many research teams studying TBI, our team was primarily composed of full-time undergraduate students or post-baccalaureate volunteers instead of graduate research assistants or clinical research coordinators. Upon successful completion of a uniform screener and consenter training, research volunteers were able to participate in recruitment of pediatric patients from the ED of a local university-affiliated pediatric hospital.

### 2.1. The Screener Role

The purpose of the screener role is to monitor ED admissions to identify candidates, rule out participants based on exclusion criteria, and communicate relevant de-identified information to personnel responsible for participant contact. During onboarding, new research volunteers completed a series of documentation and training as required by the recruitment site and research study. Volunteers were required to complete a background check, submit several documents (e.g., vaccination records, tuberculosis test, confidentiality agreement, financial interest disclosure), complete a series of trainings (e.g., Collaborative Institutional Training Initiative (CITI) Biomedical Researchers training, Health Insurance Portability and Affordability Act (HIPAA) compliance training, Good Clinical Practice training), and obtain IRB approval prior to completing a standardized screener training. The goal of the screener training is to ensure research volunteers are familiarized with the electronic health record (EHR) system of the hospital and are aware of the inclusion criteria. To fulfill the need-to-know basis requirements under HIPAA and open a medical chart, a potential case must be found to meet the following preliminary screening criteria: (1) the participant must be five to sixteen years of age, and (2) the visit reason must be indicative of possible head injury, brain injury, or concussion (e.g., MVC, fall, assault). The purpose of preliminary screening is to rule out ineligible cases without having to open the medical chart. Once a candidate was identified, the screener opened and reviewed the medical chart to confirm if the case was worth approaching. The screener also ensured that the injury was sustained within 24 h of admission into the ED, as per inclusion criteria. To ensure liability coverage in the event the hospital questioned a breach in HIPAA compliance, all screeners entered a HIPAA-compliant record of all medical charts opened using a secure webform. The screener delivered de-identified additional information (e.g., loss of consciousness, vomiting, CT scan orders) to the consenter to help determine whether the candidate ultimately met inclusion/exclusion criteria. The role of the screener is outlined in Figure 1. The role of the screener continued alongside the consenter role by providing the consenter with more information from the medical chart during consent or post-consent. The joint effort between the screener and consenter is outlined in Figure 2 and described below.

### 2.2. The Consenter Role

Consenter training is far more extensive than screening training, occurring over the span of a few weeks. In addition to the required screener onboarding materials, consenters completed the International Air Transport Association (IATA) training, which was required for transporting blood. The research volunteer was given an in-depth introduction to the purpose, details, and risks of the study to be prepared for any questions that may arise while speaking with the participant and family. Research volunteers were provided with copies of enrollment paperwork to practice thoroughly presenting its contents to the participant and LAR during the informed consent (e.g., cover letter, consent form, assent form, HIPAA form, blood draw order, documentation sheet for specimen processing). A uniform script was provided to the consenter in-training as an introductory guide to outline a sample conversation. Volunteers modified and personalized the script to better fit their approach style and individual personality. This involved the development of effective phrasing of study purpose and participant expectations, a seamless explanation of the enrollment paperwork, and a comprehensive disclosure of risks and benefits. This process was refined until the consenter in-training was comfortable with presenting the approach to experienced consenters during a mock consent. Research volunteers were also trained to administer the baseline surveys and completed a blood processing training for serum collection. Patient-Reported Outcomes Measurement Information System (PROMIS) surveys were utilized to measure variables of recovery (e.g., depressive symptoms, positive affect, physical health); surveys were hosted on REDCap, a secure clinical data management system. Finally, the consenter demonstrated their competency in a mock-environment with experienced consenters where they introduced the study, answered difficult yet realistic questions that participants and families may ask, and ensured all critical components of the study (e.g., confirmation of inclusion criteria, absence of exclusion criteria, risks, participant compensation) were addressed. Experienced consenters provided feedback regarding competencies, including the demonstration of the consenting process, professionalism, appropriate bedside manner, and blood processing. Upon training completion, the consenter was given the materials to be used for the participant approach in a consenter bag, which included a lockable HIPAA pouch, clipboard, enrollment paperwork, blood draw kits, serum collection kits, and thank you cards for clinicians, and a pen (Figure 3). Newly-approved consenters began to sign up for shifts with their first shift being a “ride-along” with an experienced consenter.

Consenters began shifts by establishing communication with the screener via text message. Consenters were encouraged to spend the duration of their shifts at the recruitment site, which minimized the amount of time between receiving a notification from the screener and the approach. Before approaching the candidate, consenters verified the injury and confirmed eligibility with the registered nurse (RN). The participant encounter was broken down into five parts: (1) a brief introduction of the study, (2) an upfront and concise disclosure of the participant expectations, (3) empathizing with the participant regarding their injury, (4) verification of inclusion criteria and absence of exclusion criteria, and (5) a thorough explanation of the enrollment paperwork to the participant and LAR. First, the consenter introduced themselves and their affiliation to the research team at the recruitment site, which demonstrated their credibility. Then, the consenter briefly explained the research study that the participant is eligible for. After introducing the study, the consenter informed the participant about what would be expected from them should they choose to enroll in the research study. As a result, participants are directly informed of expectations early on in the consent process. The consenter then empathized with the participant regarding the injury and resulting ED visit. This established rapport with the participant and allowed the consenter to better understand the injury. The consenter followed up with questions about the injury to ultimately confirm study eligibility (e.g., age, time of injury, injury severity, no prior head injury, no significant medical conditions). These preliminary steps served to gauge interest and verify eligibility, taking approximately five minutes. Finally, consenters offered to explain the enrollment paperwork in detail to the participant and LAR to provide all the information necessary to give informed consent. The enrollment paperwork outlined the logistical details of specimen collection, survey collection, a breakdown of participant compensation, and upfront risk disclosure. As the enrollment paperwork was explained, consenters encouraged participants to ask questions to clear confusion. The consent process is outlined in Figure 4. Upon written consent from the participant and the LAR, the consenter informed the RN and clinicians about participant enrollment and the need for a blood draw. The clinician, nurse practitioner, or physician assistant signed a blood draw order, and the RN performed the blood draw. The consenter then administered the baseline parental proxy and pediatric PROMIS surveys. Finally, the consenter prepared the blood sample for experimentation by isolating the serum and discarding the red blood cells.

### 2.3. Shift System

An inevitable challenge of recruiting TBI candidates is the unpredictability of when a candidate might enter the ED. Screeners and consenters utilized Google’s calendar application to create a calendar shift system. The screener/consenter added a new event to the calendar, and included their phone number and Spanish-speaking capability, if applicable, to sign up for a shift. This system allowed screeners and consenters to easily identify and contact each other during shifts. The advantages of this system included the ability to (1) sign up for different shifts each week depending on availability, (2) identify gaps and peak times in calendar coverage, and (3) maximize the range of times that consenters may approach candidates. Daily calendar coverage extended from 06:00 or earlier to 24:00 or later.

### 2.4. Analysis

The screening form responses were imported from Google Forms to Google Sheets. The form responses were then standardized and cleaned. Screening and consenting coverage was calculated in hours, by hand, using Google Calendar as the source of data. Missing values, erroneous entries, and duplicate form responses were all assessed in Google Sheets. The cleaned data were used to tally the number of individuals that advanced or were screened out at each stage of the screening and consenting process. Additionally, demographic information, ED visit reason, and other information were collected to perform analyses. Further data analyses were done through the programming language R, using linear and logistic models, correlation, and proportion statistical tests to determine the effects of screening time on enrollment and seasonal utilization of screeners. Linear and logistic regression models were used to assess statistical significance between different time periods and sub-samples of participants. Logistic regression was used specifically when determining that year was not a significant predictor of screened or consenting hours. Correlation tests were used to determine whether the year was associated with the number of hours spent screening and consenting. A chi-square goodness of fit test was not utilized, due to an incomplete data set from the Austin Census data taken from the 2018 U.S. Census Bureau [14]. Therefore, as an alternative, proportion statistical tests were used to determine statistically significant differences between race categories of the city of Austin and our enrolled participants. Two-sided *p*-values, with an alpha level of *p* = 0.05, were used as the cut-off to determine statistical significance. For the significant relationships identified, follow-up testing with one-sided *p*-values were used to identify directionality. To avoid a high probability of a Type 1 error due to multiple comparisons across racial groups, a Bonferroni correction was employed, lowering the alpha level to a more stringent criteria of *p* = 0.00833 to determine significance. Google Sheets was utilized for data visualizations.

### 2.5. Ethics Committee Approval

Approval was obtained from the University of Texas Institutional Review Board, protocol number: 2018-04-0018. All parents or LARs of candidates 5–16 years old were provided written informed consent prior to the administration of surveys. All candidates ages 7–16 years old were provided written informed assent.

## 3. Results

There were 5008 screening form entries. According to Figure 5, 20.6% of those were screened out as duplicate records (*n* = 1033). 79.37% of the unique cases were screened for the study and recorded in the screening form (*n* = 3975). As seen in Figure 5, 73.31% of those cases met preliminary criteria, allowing for the chart to be opened and for the candidate to be further screened (*n* = 2914). A total of 8.65% of those that fully qualified were approached by consenters (*n* = 252). Of those approached, 36.91% enrolled in the study; 33 as head injury cases and 60 as orthopedic controls (*n* = 93).

During the first stage of screening, 26.69% of the screened individuals for possible enrollment into the study did not meet preliminary criteria (*n* = 1061). As seen in Figure 5, 73.79% of the candidates did not meet the age criteria after Stage 1 (*n* = 783). During the second stage of screening, of the 2914 cases that were further screened, 42.69% were in the emergency department for another reason; other excluders can be found in Figure 5 (*n* = 1244). Within the third stage of screening, consenters approached candidates, 63.09% of which were not enrolled (*n* = 159). Specifically, 22.01% of candidates were deemed ineligible during the consent approach due to the identification of exclusion criteria (*n* = 35). Of the remaining eligible individuals, 25.78% declined because of the blood draw; other reasons are listed in Figure 5 (*n* = 41).

Of the 3975 screened individuals, 19.87% did not meet the study’s age criterion as seen in Figure 5 (*n* = 790). At stage 1, 1061 candidates were excluded; 68.24% were excluded for being under the age criteria, while 5.56% were excluded for being over the age criteria. During stage 2, 0.34% were excluded due to age. Table 1a displays a detailed breakdown of the ages enrolled and screened.

As seen in Table 1a, 11.3% and 40.9% of the total screened and enrolled, respectively, were female. Comparison of the 2018 Austin population demographics to the participant demographics showed no significant difference in sex (*p* = 0.1305), with 49.3% of the Austin population being female, and 40.9% of the participants being female (*n* = 38). A total of 34.30% of the Austin population and 50.5% of the participants (*n* = 46) were Hispanic. The proportion of Hispanics was significantly greater in the participants than in the Austin population (*p* = 0.002106). Table 1b compares the proportions of race categories in the Austin population to those of participants. The demographic profile for race was significantly different between the Austin population and participants (*p* = 0.000102). Specifically, the proportion of participants who reported Hawaiian/Pacific Islander as their race was significantly greater than that of Austin residents (*p* = 0.00000238). All other categories of race were not significantly different between the 2018 Austin population and participants as seen in Table 1b.

The highest enrollment occurred during September 2018 (*n* = 18). Concomitantly, there was a significant increase in screening coverage from August 2018 to September 2018 (*p* = 0.0043). A total of 1249 charts were opened during the months of June to December in 2018, followed by 2725 opened charts during the months of January to November in 2019. In 2018, a total of 1936.3 h were spent screening compared to 4663.3 h in 2019. The monthly enrollment rate for 2018 averaged at 8.83 participants per month, which was significantly higher than the 3.45 participants per month in 2019 (*p* = 0.0462). The total number of participants that consented per month is seen in further detail in Figure 6.

One day of screening was defined to be 24 h. From July 2018 to December 2018, 44.02% of a day was spent screening, on average. From January 2019 to November 2019, 57.86% of a day was spent screening, on average. In total, 1936.3 h were spent screening charts from July 2018 to December 2018, and a total of 4663.25 h were spent screening charts from January 2019 to November 2019. The total hours spent screening charts differed significantly (*p* = 0.047) between the years 2018 and 2019. Figure 6 displays the total hours spent screening per month.

At stage 1 of screening, three candidates were excluded from the study due to absent protocol for enrolling Spanish-speaking participants, as per Figure 5. Further screening of the EHR during stage 2 excluded 28 Spanish-speaking candidates, as per Figure 5. During the consent approach, five additional candidates were excluded, as per Figure 5. A total of 22.54% of the research team reported proficiency in Spanish. There was a significant difference in self-identification of Hispanic ethnicity between participants and research team members, as seen in Table 2 (*p* = 1.04 × 10^−8^).

From July 2018 to November 2019, the research team consisted of 71 members. There was a significant difference in racial representation between White team members and participants (*p* = 4.20 × 10^−16^) and Asian team members and participants (*p* = 2.20 × 10^−16^). There were no other significant differences observed in race. Within the research team, 71.8% were female (*n* = 51) and 85.9% were full-time undergraduate students (*n* = 61).

## 4. Discussion

### 4.1. Screening Out at Different Stages of Enrollment

Initially, there were 5008 screening form entries, 20.63% of which were duplicate responses (*n* = 1033). The large number of duplicates found in screening is accounted for by an initially less organized screening process due to shift overlap and insufficient communication. Limiting the number of duplicates is critical to ensuring the security of protected health information (PHI), such that it is not repetitively accessed by multiple screeners. As the screening process evolved, communication between screeners during and between their respective screening shifts improved using Slack. One change that has been made to limit the number of duplicates was the use of an “on-call” channel in Slack. This communication channel allows screeners and consenters to communicate with others during their shift and reference previous discussions to avoid unnecessarily opening charts. In addition, the research team implemented a new screening schedule, ensuring a maximum of two screeners at a time while minimizing shift overlap. Due to inefficient data collection methods, new screening and consenting forms are being developed in an attempt to increase data lineage. A temporary change to the current form allows screeners to indicate when a screener other than themselves will be responsible for completing the form. This allows the screener to finish screening a participant that spans multiple shifts.

### 4.2. Demographics of Enrolled Patients vs. Census

When comparing demographic statistics for the percentage enrolled against the percentage screened, various factors including age, sex, ethnicity, and race were considered.

Demographics of participants were compared to the Austin population based on the 2018 U.S. Census Bureau. When considering sex, according to Table 1, 49.3% of Austin’s population and 40.86% of participants were female [14]. There was no significant difference found between these two populations, indicating that the sex demographics of the study are representative of the Austin population. Typically, the national incidence of female TBI is considerably low; however, female TBI enrollment in our study is high, thus adequately representing female TBI participants [15]. In terms of ethnicity, 65.7% of the Austin population and 49.46% of the participants were Non-Hispanic. There was a significant difference between the two populations; the proportion of Hispanic participants was overrepresented in comparison to the Austin population (*p* = 0.0021). This increased Hispanic representation results in a more heterogeneous pool of participants. When comparing race demographics of participants to the Austin population, as seen in Table 1, there were no significant differences seen in most categories, except for the Hawaiian/Pacific Islander category. The number of enrolled participants who identified as Hawaiian/Pacific Islander was significantly higher than those represented in the Austin population (*p* = 0.00000238). Overall, results suggest that the participants accurately represented the Austin population; for some racial groups, the participants had greater representation than in the Austin census data.

### 4.3. Screening and Enrollment Rates

Screening for candidates is a vital role in the process of consenting since screeners are responsible for identifying candidates. There was a 118% increase in charts opened in 2019 compared to 2018. Additionally, the total hours spent screening quadrupled from 2018 to 2019, which is significantly different (*p* = 0.047). A significant increase in screening can result in a higher likelihood of enrollment. Increased screening coverage between August 2018 and September 2018 resulted in the most successful month of enrollment, with 18 consenting participants. Overall, the enrollment rate in 2018 was significantly higher than in 2019. The coefficient of determination was calculated to be 0.24, showing low predictability value; however, it is important to note that the enrollment rates obtained are not for extrapolation of future enrollment rates.

### 4.4. Spanish-Speaking Enrollment

Participant recruitment took place in a pediatric hospital with a large Spanish-speaking population. In January 2019, our research team identified a gap in participant enrollment, as we lacked the resources to enroll participants who did not speak English. Initially, when a candidate was a Spanish speaker, either a Spanish-speaking consenter was contacted to approach the candidate or a consenter used a virtual translator. However, the former proved to be logistically difficult and inefficient, and the latter often led to imprecise interpretation and inadequate communication during risk disclosure. Our team addressed this by translating enrollment paperwork (e.g., cover letter, consent, assent, HIPAA form), saliva collection instructions, and post-consent surveys. Once these documents were translated and approved by the IRB, research volunteers began formally consenting Spanish-speaking families as of August 2019.

As per Figure 6, the lower enrollment rate in 2019 may be partly due to a pause in enrollment of controls. In addition, changes to consenter training in Fall 2019 led to fewer consenters present on-site. This contributed to delays (e.g., increased transportation time) in the consent process. The consenting approach was not standardized, leading to variation in explanation of the study and its risks to candidates and the LAR. The research team has since increased accessibility and standardized the consenter training by producing online modules through Google Classroom.

A total of 33 candidates were excluded from the study across all stages of the enrollment process due to absent protocol to enroll Spanish-speaking candidates. To combat this, Spanish participant surveys were obtained from PROMIS and enrollment paperwork was translated into Spanish. With the help of Spanish-speaking team members, training for bilingual consenters was initiated. Beginning in September 2019, consenters were re-trained and taught basic Spanish to feel comfortable initiating the consent process even if they were not bilingual. Therefore, any consenter fluent in at least the introductory script would be equipped to gauge candidates’ interest while the consenter arranges for an official translator or a Spanish-speaking team member.

### 4.5. Demographics of Research Team Members Compared to Enrolled Participants

Traditional clinical research studies rely heavily on patients’ physicians and full-time clinical staff for enrollment, while effective concerns have been raised about if the established rapport between patients and clinical staff affects a patient’s decision to consent [16]. In addition, Bell et al. [16] acknowledge clinicians often lack the requisite knowledge and time to fully commit to the recruitment process. To address this concern, Bell et al. recommend bringing on additional personnel to assist with recruitment and acknowledges that reliance on volunteers requires additional training and oversight. Our new modality incorporates carefully selected and trained individuals to screen and consent candidates during recruitment. By having research volunteers solely dedicated to screening EHRs in search of study participants, efficiency and representation of the population in recruitment can be improved.

Team members speak a variety of other languages and come from diverse places around the world, as seen in Table 2. Diversity is important to increase cultural competency of consenters, which helps establish rapport with candidates, increasing comfort and enthusiasm about participating in the study.

Notably, although our team has a large proportion of Asian team members, the study sample had an underrepresentation of Asian participants (7.96%); however, this enrollment is representative of the Austin, Texas population (7.3% Asian) from Census.gov. Moreover, lower rates of enrollment in this study are consistent with lower rates of hospitalization among Asian children. A CDC report [17] found that the rates of TBI-related hospitalizations (per 100,000) in the Asian, American Indian, Alaska Native, or Pacific Islander pediatric population aged 5–14 (ages 5–9, 12.9; ages 10–14, 32.2) were less than the rates of hospitalizations in the Black pediatric population (ages 5–9, 35.2; ages 10–14, 47.6) and White pediatric population (ages 5–9, 35.3; ages 10–14, 38.6). There is a need to further explore other factors that may lead to both the decreased proportion of Asian participants and decreased hospitalization for pediatric TBIs in the Asian population.

## 5. Conclusions

The goal of this study is to describe a student-led, participant-focused consenting strategy supporting the enrollment of more diverse populations into pediatric TBI biobanks. While the screener role and consenter role entail separate responsibilities, the interplay of both is crucial for participant recruitment. We found that strategies such as specifically training certain consenters to work with Spanish-speaking participants helped influence the diversity and efficiency of enrollment. Additionally, strategies such as the implementation of “ride-alongs” achieved the same goal by streamlining the consenter training process. These preliminary findings highlight the importance of employing effective communication strategies to widen screener and consenter coverage and maximize enrollment in diverse populations. A multi-staged screening and consent modality, in combination with slight modifications such as the addition of a bilingual consent strategy, may improve enrollment outcomes. Full-time research coordinators are not necessary for effective enrollment; rather, a student-led team is sufficient for developing an adequate enrollment strategy. Screening coverage can be maximized by utilizing available technologies to create a scheduling system for shifts and consenting coverage can be maximized by employing a ride-along system for newly trained consenters.

The research team may modify the standard approach as necessary to better represent and support the recruitment site, research team, and local demographics. For example, a large Spanish-speaking population suggests the need for Spanish-translated enrollment paperwork. An understanding of local demographics is important for increasing overall cultural competency within the research team; this in turn, can build rapport with candidates and familiarity with a bilingual consent approach. Since cultural differences seem to influence participant expectations of the research process [15], a diversity-sensitive consent modality may be key to expanding the biobank and better understanding pediatric TBI.

## Figures and Tables

**Figure 1 brainsci-10-00434-f001:**
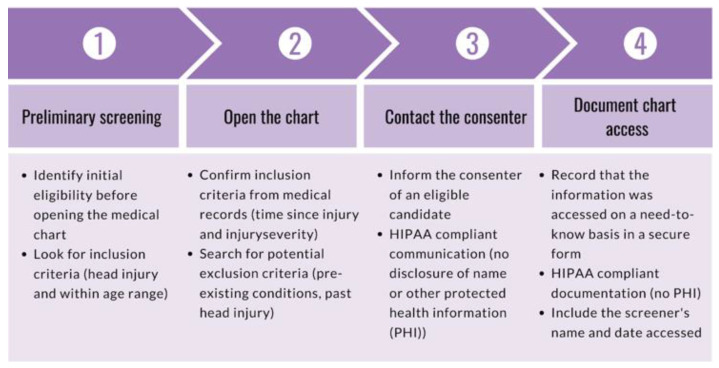
Diagram outlining 4-part screening process: (1) preliminary screening, (2) chart opening, (3) screener-consenter communication, and (4) documentation of chart access.

**Figure 2 brainsci-10-00434-f002:**
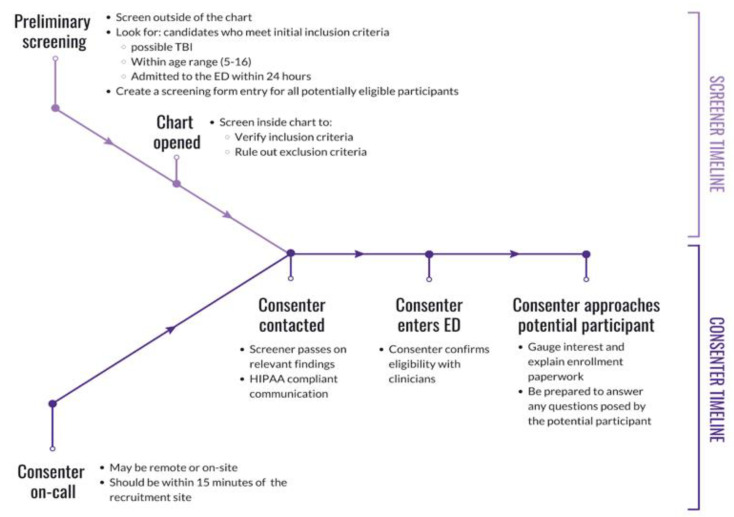
Diagram outlining the interaction between the screener role and consenter role in the recruitment process.

**Figure 3 brainsci-10-00434-f003:**
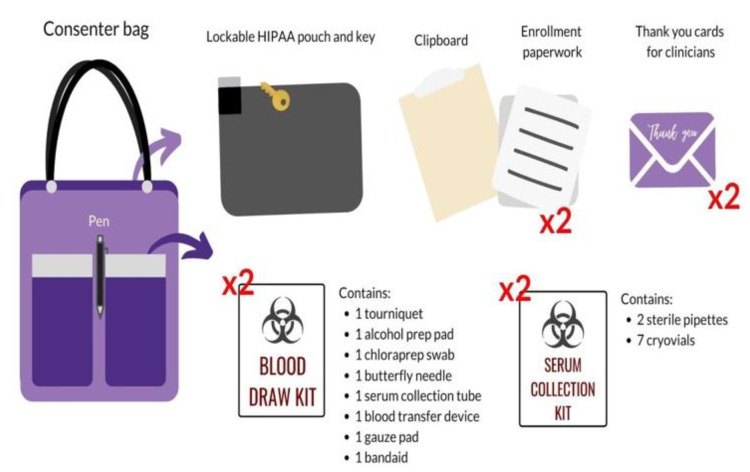
The contents of a consenter’s bag include a lockable HIPAA pouch and key, a clipboard, two copies of enrollment paperwork, a laptop for digital data collection, 2 blood draw kits, 2 serum collection kits, 2 clinician thank you cards, and a pen. Preparing duplicate sets allows for the enrollment of two participants in one shift if necessary.

**Figure 4 brainsci-10-00434-f004:**
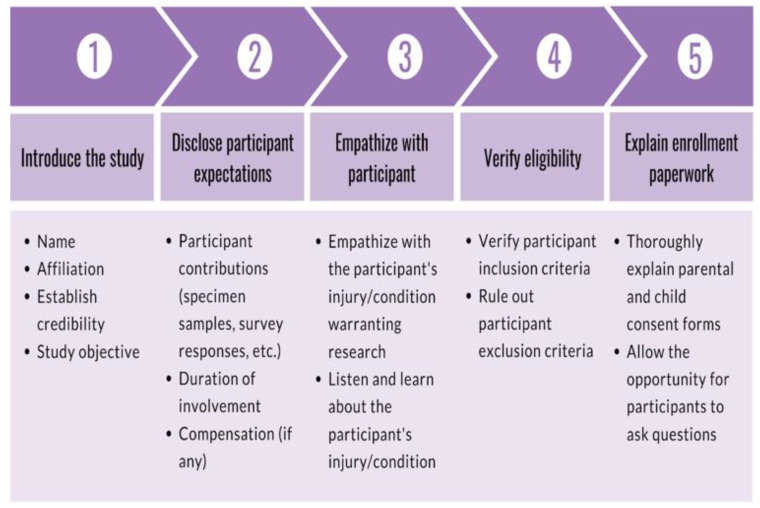
Diagram outlining 5-part consent process: (1) a brief introduction of the study, (2) an upfront and concise disclosure of the participant expectations, (3) empathizing with the patient regarding their injury, (4) verification of inclusion criteria and absence of exclusion criteria, and (5) a thorough explanation of the enrollment paperwork to the patient and LAR.

**Figure 5 brainsci-10-00434-f005:**
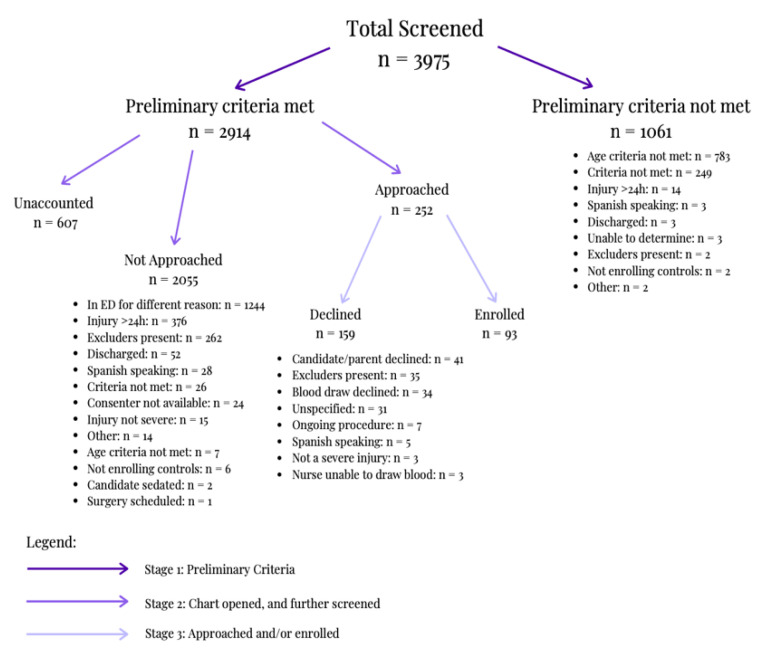
Breakdown of unique cases screened and recorded in screening form.

**Figure 6 brainsci-10-00434-f006:**
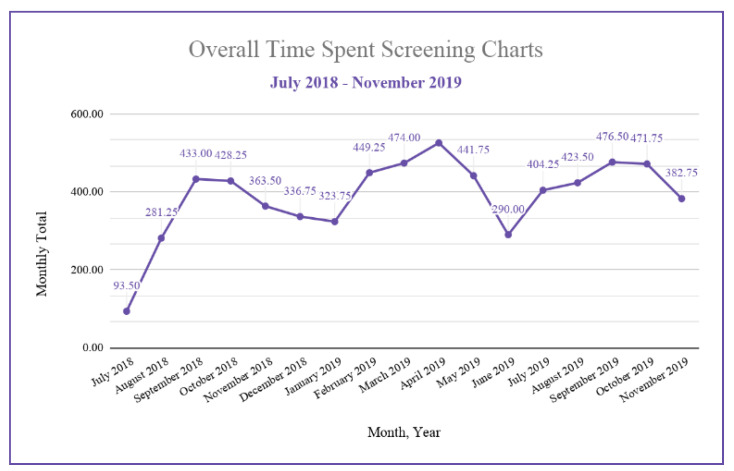
Overall hours spent screening charts from July 2018 to November 2019.

**Table 1 brainsci-10-00434-t001:** Demographics of screened and enrolled vs. Austin population.

**a. Demographics of Screened and Enrolled.**					
**Demographic Measure**		**Enrolled Frequency (*n* = 93)**	**Percent**	**Screened Participants (*n* = 3975)**	**Percent**
Age					
	<5	-	-	781	19.64%
	5–8	19	20.43%	974	24.50%
	9–12	41	44.09%	1172	29.48%
	13–16	33	35.48%	980	24.65%
	>16	-		68	1.71%
**b. Enrolled Demographics compared to City of Austin Census Demographics**					
**Demographic Measure**		**Enrolled Participants (*n* = 93)**	**City of Austin Population ^1^ (*n* = 964,243)**	***p***	***p*, Post-Hoc**
Sex		40.86%	49.30%	0.1035	
Ethnicity					
	Hispanic	50.54%	34.30%	0.0042 **	0.0021 ^✣✣^
	Non-Hispanic	49.46%	65.70%	0.0042 **	0.0021 ^✣✣^
Race ^2^				0.000102 **	
	White	74.19%	73.47%		0.874
Black or African American		10.75%	8.85%		0.085
American Indian/Alaskan Native		1.08%	0.92%		0.875
	Asian	2.15%	7.96%		0.038
Hawaiian/Pacific Islander		2.08%	0.047%		2.38 × 10^−6 ✣✣✣^
	Other	10.75%	8.75%		0.495

Notes. * *p* < 0.05, two tailed. ** *p* < 0.01, two tailed; ^✢^
*p* < 0.05, one tailed. ^✢✢^
*p* < 0.01, one tailed. ^✣✣✣^
*p* < 0.001, one tailed. ^1^ City of Austin Population estimate as of 2018, according to the United States Census Bureau. ^2^ Post-hoc *p*-values calculated with Bonferroni correction, new alpha level *p* = 0.00833.

**Table 2 brainsci-10-00434-t002:** Research team demographics compared to enrolled participants based on sex, ethnicity, and race.

Lab Members Demographics Compared to Enrolled Participants—Sex, Ethnicity, and Race.					
Demographic Measure		Lab Members (*n* = 71)	Enrolled Participants (*n* = 93)	*p*	*p*, Post-Hoc ^1^
Sex		71.83%	40.86%	1.104 × 10^−7^ ***	5.518 × 10^−8 ✣✣✣^
Ethnicity					
	Hispanic	15.49%	34.30%	1.04 × 10^−8^ ***	5.185 × 10^−9 ✣✣✣^
	Non-Hispanic	84.50%	65.70%	1.037 × 10^−8^ ***	5.185 × 10^−9 ✣✣✣^
Race				2.20 × 10^−16^ ***	
	White	32.39%	73.47%	8.41 × 10^−16^ ***	4.20 × 10^−16 ✣✣✣^
Black or African American		2.82%	8.85%	0.0392	
American Indian/Alaskan Native		0%	0.92%	0.0392	
	Asian	57.75%	7.96%	2.20 × 10^−16^ ***	2.20 × 10^−16 ✣✣✣^
Hawaiian/Pacific Islander		0%	0.047%	0.0392	
Other/More than one		1.40%	8.75%	0.01105	
	Unknown	1.40%	0.00%	-	-
Secondary Language					
	Spanish	22.54%	-	-	-
	Other ^2^	46.41%	-	-	-
Education Classification					
	Undergraduates	85.92%	-	-	-
Graduates/Post-Baccalaureate		14.08%	-	-	-

Notes. * *p* < 0.05, two tailed. ** *p* < 0.01, two tailed, *** *p* < 0.001, two tailed; ^✢^
*p* < 0.05, one tailed. ^✣✣^
*p* < 0.01, one tailed, ^✣✣✣^
*p* < 0.001, one tailed. ^1^ Post-hoc *p*-values calculated with Bonferroni correction. ^2^ Other languages include: Arabic, Farsi, French, Haitian, Hindi, Kannada, Korean, Malayalam, Marathi, Punjabi, Russian, Sanskrit, Tagalog, Tamil, Telugu, Turkish, Urdu, Vietnamese.

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
