# Peer review of "A Novel Clinical Research Modality for Enrolling Diverse Participants Using a Diverse Team"

_brainsci, 2020, doi:10.3390/brainsci10070434_

Round 1
Reviewer 1 Report
Lay and colleagues present a manuscript describing a novel clinical research modality to establish a practice that aims to increase the enrollment of a diverse pediatric TBI population for enrollment in biobank studies. The study is an important contribution to the field that will help us increase the diversity of pediatric TBI participants in clinical studies. I believe the paper is acceptable for publication in its current form. However, I would ask that the authors add some information in the discussion about how their enrollment process and screener and consenter roles differ from the current methods traditionally used in the enrollment process.
Reviewer 2 Report
Building a diverse student team and having undergraduate students to recruit patients is a cost-effecient way if it works well. The authors have described an interesting way to screen patients and the special processes of training consenters. These processes result in recruiting more hispanic patients. A few comments for the authors
1. The section describing data analyses needs improvement. It is not clear which method are used for which type of analyses or variables. For instance, logistic regression were used for anaylyzing what? analyzing categorical variables when comparing xxx? Linear model were used for what?
2. Result.
a. When present the results in tables, it is not necessary to present both one-side and two-side p-values, two-side p-values are sufficent. Otherwise, it is confusing to see so many P-values
b. In Table 1b, there were significant difference in Asian cateogy, between the enrolled and city of Austin, you did not comment these in your results.
3. Discussion.
Your research team has more than 50% Asians, why you have recruited even less Asian patients then the census (2.15 vs 7.96%), you need to explain and may list as a limitation.
Reviewer 3 Report
In this manuscript, the authors aim to introduce a novel clinical research modality to improve the practices for recruiting a diverse pediatric TBI population and quantifying the key information on enrollment into biobanks.
Overall, the study was well designed, the methods and analysis were clearly described, the results are convincing and informative.
